# Clinical Performance of Self-Collected Nasal Swabs and Antigen Rapid Tests for SARS-CoV-2 Detection in Resource-Poor Settings

**DOI:** 10.3390/biomedicines10092327

**Published:** 2022-09-19

**Authors:** Nádia Sitoe, Júlia Sambo, Nédio Mabunda, Neuza Nguenha, Jorfélia Chilaúle, Júlio Rafael, Anésio Macicame, Imelda Chelene, Chishamiso Mudenyanga, Jillian Sacks, Sofia Viegas, Osvaldo Loquiha, Ilesh Jani

**Affiliations:** 1Instituto Nacional de Saúde, Marracuene 3943, Mozambique; 2Clinton Health Access Initiative, Maputo City 592, Mozambique; 3World Health Organization, 1211 Geneva, Switzerland

**Keywords:** nasal swabs, self-collection, antigen rapid tests, clinical evaluation, resource-poor settings

## Abstract

**Background:** In resource-poor countries, antigen-based rapid tests (Ag-RDTs) performed at primary healthcare and community settings improved access to SARS-CoV-2 diagnostics. However, the technical skills and biosafety requirements inherent to nasopharyngeal and oropharyngeal (OP) specimens limit the scale-up of SARS-CoV-2 testing. The collection of nasal-swabs is programmatically viable, but its performance has not been evaluated in resource-poor settings. **Methods:** We first evaluated the performance of SteriPack self-collected nasal swabs for the detection of SARS-CoV-2 by real-time PCR in 1498 consecutively enrolled patients with suspected infection. Next, we evaluated the clinical performance of three nasal swab-based Ag-RDTs against real-time PCR on OP specimens. **Results:** The sensitivity of nasal swabs was 80.6% [95% CI: 75.3–85.2%] compared to OP specimens. There was a good correlation (r = 0.58; *p* < 0.0001) between Ct values of 213 positive cases obtained using nasal and OP swabs. Our findings show sensitivities of 79.7% [95% CI: 73.3–85.1%] for Panbio COVID-19 Ag-RDT, 59.6% [95% CI: 55.2–63.8%] for COVIOS Ag-RDT, and 78.0% [95% CI: 73.5–82.0%] for the LumiraDx SARS-CoV-2 Ag-RDT. **Conclusions:** In our setting, the COVIOS Ag-RDT did not meet WHO requirements. Nasal swab-based Ag-RDTs for SARS-CoV-2 detection constitute a viable and accurate diagnostic option in resource-poor settings.

## 1. Introduction

Accurate and rapid diagnosis remains a key strategy to control the COVID-19 pandemic [1,2]. In resource-limited settings, the impact of diagnosis on reducing the spread of SARS-CoV-2 is reliant on wide access to testing in primary healthcare and community settings [1,3,4]. The deployment of antigen rapid diagnostic tests (Ag-RDTs) for SARS-CoV-2 dramatically changed the availability of testing [5] during the second year of the pandemic, and for the first time in resource-limited settings, allowed the timely diagnosis of SARS-CoV-2 infection [6] outside the main urban centers.

Most SARS-CoV-2 Ag-RDTs require the use of nasopharyngeal or oropharyngeal specimens [7,8,9]. The collection of these biological samples requires invasive procedures, trained personnel, and complex biosafety protocols [10,11,12]. These factors can hamper the scale-up of testing as a public health strategy due to operational challenges and high programmatic costs [10,11,12].

Nasal swabs constitute viable biological specimens for SARS-CoV-2 testing, for both RNA amplification and antigen detection [13,14]. The collection of swabs from nostrils is easier to perform, does not involve aerosol-generating [15] procedures, and is suitable to be conducted by lay personnel [16]. Therefore, the adoption of nasal swab-based SARS-CoV-2 diagnosis can be catalytical for the implementation of mass testing in resource-limited settings [10,11].

We first evaluated the performance of nasal swabs as an alternative sample for detecting SARS-CoV-2 by real-time PCR. Subsequently, we determined the diagnostic performance of nasal swab-based Ag-RDTs under health system conditions in Mozambique.

## 2. Materials and Methods

### 2.1. Study Setting and Participants

Cross sectional prospective studies were conducted between April 2021 and January 2022. Study participants were recruited at five public health facilities in the Maputo area, namely Centro de Saúde de Marracuene (CSM), Hospital Provincial da Matola (HPM), Hospital Geral de Chamanculo (HGC), Hospital Geral de Mavalane (HGM), and Hospital Central de Maputo (HCM).

Participants aged 18 years old and above suspected of SARS-CoV-2 infection and contacts of confirmed cases were enrolled in the study. Definitions of suspected cases and their contacts were based on the World Health Organization (WHO) guidelines [9].

An electronic standardized case investigation form (CIF) was completed for each participant to collect the demographic, clinical, epidemiological, and testing data. Electronic forms in mobile tablets (Samsung, Suwon-si, South Korea, model: TAB A 8”) were used for this purpose. The quality of data entered was randomly verified as part of quality control procedures.

### 2.2. Sample Collection

For the evaluation of the self-collected Sterile Polyester Nasal Swabs (SteriPack, Lakeland, FL, USA, REF. 60,564 RevA), the technology under evaluation, each participant self-collected one nasal sample by inserting the swab into both nostrils until the level of the turbinates. Participants were instructed to rotate the swab a few times against the nasal wall in each nostril. Subsequently, study staff collected an oropharyngeal swab that was used for the routine diagnosis.

Nasal and oropharyngeal swabs were placed into sterile tubes containing 3 mL of viral transport media (iClean, Shenzhen, China, REF. CY-F005-20) with antibiotics (Gentamicin, Streptomycin, Penicillin, Amphotericin B), transported and stored at 2–8 °C in the testing laboratory for a maximum of 24 h before testing. All samples were referred for RT-PCR testing at the Instituto Nacional de Saúde. Testing of paired swabs was blinded to laboratory technicians performing the assay and only results obtained on oropharyngeal swabs were issued to participants.

For the evaluation of nasal swab-based Ag-RDTs, the collection of nasal swabs for Ag-RDT testing (technologies under evaluation) and the oropharyngeal swab for RT-PCR testing (gold standard) were performed by the study staff. Study staff performed the Ag-RDTs at health facilities, while oropharyngeal swabs were referred to the Instituto Nacional de Saúde for RT-PCR by blinded laboratory technicians.

### 2.3. Sample Size

Sample size was determined considering a minimum positivity rate for SARS-CoV-2 of 4.5%, 95% sensitivity and specificity, and 95% confidence interval to detect at least 85 positive cases in each evaluation, among the symptomatic cases and their contacts. A total of 1498 subjects (269 positive cases) consecutively participated in the evaluation of nasal swabs samples for RT-PCR: 696 subjects (386 positive cases) in the LumiraDx SARS-CoV-2 Ag Test, 1123 subjects (520 positive cases) participated in COVIOS Ag Rapid Test Device, and 400 subjects (214 positive cases) participated in Panbio COVID-19 Ag evaluations.

### 2.4. Antigen Rapid Test (Ag-RDT) Testing

Three nasal swab-based SARS-CoV-2 Ag-RDTs were evaluated, namely: (i) Panbio^TM^ COVID-19 Ag rapid diagnostic test device nasal (Abbott, Jena, Germany, Ref: 41FK11); (ii) COVIOS^®^ Ag COVID-19 Rapid Antigen Test (Global Access Diagnostics, United Kingdom, Ref: 11811125, Lot: CA25K-121-2); and (iii) LumiraDx SARS-CoV-2 Ag Test (LumiraDx, London, UK, Ref.: L016000109048, Lot.: GM2000390). Each test was performed according to the manufacturer’s instructions. The LumiraDx test is an automated process assay while the other two are manual tests. The samples for all evaluated tests were collected according to specifications from each manufacturer.

Panbio^TM^ COVID-19 Ag rapid diagnostic test: A nasal swab with the specimen was inserted into an extraction buffer tube provided by the manufacturer. The swab was swirled at least five times. The swab was removed from the tube squeezing the sides to extract absorbed liquid. Three to five drops of the liquid in the tube were applied to the specimen well of the test device and the test result was read between 15 and 20 min according to the manufacturer’s procedures. The limit of detection of this Ag-RDT was 2.5 × 10^1.8^ TCID_50_/mL of SARS-CoV-2.

COVIOS^®^ Ag COVID-19 Rapid Antigen Test: A nasal swab was collected and inserted into an extraction buffer tube provided by the manufacturer. The swab was stirred at least five to ten times, and then removed while squeezing the sides to extract liquid from swab. Five drops were applied to the test device and the result was read in 10 min. The limit of detection of this Ag-RDT was 3.5 × 10^2^ TCID_50_/mL of SARS-CoV-2.

LumiraDx SARS-CoV-2 antigen test: A nasal swab was collected and then placed into 0.7 mL of extraction buffer. One whole drop of the sample was then applied onto the Test Strip Sample Application Area when prompted by the instrument. The result was read by the LumiraDx equipment (LumiraDx, London, United Kingdom) within 12 min. The results appeared on the touchscreen as Negative (–) SARS-CoV-2, Positive (+) SARS-CoV-2, or invalid results. The limit of detection of this Ag-RDT was 2.5 × 10^5^ TCID_50_/_mL_ of SARS-CoV-2.

### 2.5. Real-Time PCR Testing

Automated SARS-CoV-2 RNA extraction and detection were executed on the Abbott m2000 sp/rt platforms (Abbott Molecular Inc., Taipei city, Taiwan). The RT-PCR assay uses two sets of primers to amplify regions within the highly conserved RNA-dependent RNA polymerase (RdRp) and N genes. The limit of detection of the kit was 100 copies/mL of SARS-CoV-2. Results were considered positive if both genes were detected. If only one gene was detected, then the result was considered indeterminate. If this occurred, an additional oropharyngeal specimen collection was requested to ensure that the participants had their test results, but the final result was not included in the evaluation.

### 2.6. Data Analysis

Results generated by the evaluated technologies were compared to those yielded by oropharyngeal swabs tested by RT-PCR. The performance of self-collected nasal swabs and Ag-RDTs was determined by their sensitivity, specificity, positive and negative predictive values [17], observed agreement, and kappa agreement. Wilcoxon signed rank tests for paired samples were used to compare the Ct values obtained in self-collected nasal and oropharyngeal swabs. The Spearman coefficient determined the correlation and *p* < 0.05 was considered significant. Statistical analysis was performed using Microsoft Excel 360 (Microsoft Co., Redmond, WA, USA).

## 3. Results

### 3.1. Characteristics of Study Population

A total of 3717 participants were enrolled in the evaluations, of which 54.3% (2017/3717) were female (Table 1). The median age of participants was 35 years (Min-Max: 18–91 years). The overall SARS-CoV-2 positivity rate by RT-PCR was 37.6%. Most of the participants (3443/3717; 92.6%) were symptomatic and 83.1% (3089/3717) reported having symptoms for less than seven days. The median Ct value in oropharyngeal swabs was 24.7 (Min-Max: 5.7–36.5) and 25.7 (Min-Max: 3.7–39.0) during the evaluations of nasal swabs and Ag-RDTs, respectively.

### 3.2. Performance of Nasal Swabs for SARS-CoV-2 Detection

We first evaluated the performance of self-collected SteriPack nasal swabs when compared to oropharyngeal swabs in 1498 patients with suspected SARS-CoV-2 infection (Table 2). The overall sensitivity, specificity, positive predictive value (PPV), and negative predictive value (NPV) were 80.6% (95% CI: 75.3–85.2%), 96.4% (95% CI: 95.2–97.4%), 83.1% (95% CI: 78.0–87.4%), and 95.8% (95% CI: 94.5–96.8%), respectively. The observed result agreement between the two types of swabs was 93.5% (95% CI: 34.7–94.7%). Among the 213 patients with SARS-CoV-2 positive results on paired swabs, there was good correlation (r = 0.58; *p* < 0.0001) of the amount of virus detected as measured by the Ct value (Figure 1). The median Ct value in the evaluation of nasal swabs was 22.41 (Min-Max: 4.4–36.9) for self-collected nasal swabs and 23.67 (Min-Max: 5.7–34.7) for oropharyngeal swabs (*p* = 0.068).

### 3.3. Performance of Nasal Swab-Based Antigen Rapid Tests for SARS-CoV-2 Detection

Having shown good performance of nasal swabs for detecting SARS-CoV-2 genetic material, we next assessed the clinical performance of three nasal swab-based SARS-CoV-2 Ag-RDTs (Table 2). The overall sensitivity, specificity, PPV, and NPV observed for the Panbio COVID-19 Ag Rapid Test Device were 79.7% (95% CI: 73.3–85.1%), 97.2% (95% CI: 93.5–99.1%), 96.8% (95% CI: 92.8–99.0%), and 81.4% (95% CI: 75.5–86.4%), respectively. The agreement observed between results generated by the Panbio device and those yielded by RT-PCR was 88.0% (95% CI: 41.9–91.2%). The overall Cohen’s kappa was 0.71 (95% CI: 0.70–0.72). The performance of the nasal swabs was similar in both symptomatic and non-symptomatic individuals.

The overall sensitivity, specificity, PPV, and NPV observed for the COVIOS Ag COVID-19 Rapid Antigen Test were 59.6% (95% CI: 55.2–63.8%), 94.9% (95% CI: 92.8–96.5%), 91.1% (95% CI: 87.6–93.9%), and 72.6% (95% CI: 69.3–75.8%), respectively. The agreement observed between results generated by the COVIOS device and those yielded by RT-PCR was 78.3% (95% CI: 37.0–80.7%). The overall Cohen’s kappa was 0.556 (95% CI: 0.551–0.560). There were few positive individuals without symptoms in the evaluation of this test (*n* = 8), which did not allow for meaningful comparison of the performance of the test according to presence or absence of symptoms.

For the LumiraDx SARS-CoV-2 Ag Test, the overall sensitivity, specificity, PPV, and NPV observed were 78.0% (95% CI:73.5–82.0%), 95.8% (95% CI: 92.9–97.7%), 95.9% (95% CI: 93.0–97.8%), and 77.6% (95% CI: 73.0–81.7%), respectively. The agreement observed between results generated by the LumiraDx device and those yielded by RT-PCR was 85.9% (95% CI: 43.7–88.4%). The overall Cohen’s kappa was 0.72 (95% CI: 0.72–0.73). The performance of the nasal swabs was similar in both symptomatic and non-symptomatic individuals. The sensitivity of the LumiraDx test was higher among symptomatic (79.9%; 95% CI: 75.4–83.9%) than among non-symptomatic individuals (47.8%; 95% CI:26.8–69.4%), with Cohen’s kappa of 0.71 (95% CI: 0.70–0.72) and 0.50 (95% CI: 0.36–0.65), respectively.

## 4. Discussion

During the onset of the COVID-19 pandemic, PCR-based assays performed on oropharyngeal and nasopharyngeal swabs were the only option available to diagnose SARS-CoV-2 infections [9,18,19], severely limiting the availability of testing in LMICs. Although the later arrival of Ag-RDTs increased the volume of testing in these countries, the need for oropharyngeal and nasopharyngeal swabs continued to pose programmatic challenges [15].

In this study, we showed that nasal swabs constitute an accurate alternative to oropharyngeal swabs for the detection of SARS-CoV-2 in the upper respiratory tract. Although the sensitivity of nasal swabs was lower than the gold standard, the method met the WHO requirements [20], and results were in line with the sensitivities of 82–88% reported in a recent meta-analysis [21]. We performed the evaluation during the third and fourth waves of SARS-CoV-2 in Southern Africa and in a mostly unvaccinated population. This epidemiological context of our study setting and population immunity resulting from previous infections can justify the lower sensitivity observed. It has been reported that the sensitivity of diagnostic tests can be negatively affected by lower concentrations of the virus [22] in nasal secretions. Additionally, it is documented that immunity from previous infections reduces the replication of SARS-CoV-2 in the nasal mucosa [23]. It has been suggested that combining nasal swabs and saliva collection may increase the sensitivity of SARS-CoV-2 detection [13], however this is impractical in real-life settings.

After demonstrating good accuracy of nasal swabs for detecting SARS-CoV-2 genetic material, we evaluated three Ag-RDTs based on nasal swabs. Two of these, Panbio and LumiraDx, met the WHO requirements for selection of essential in vitro diagnostics for SARS-CoV-2 [20]. Nevertheless, the sensitivities documented here were lower than the >90% reported by the manufacturers. The third assay evaluated, *COVIOS^®^ Ag COVID-19 RDT,* had a sensitivity of 59.6%, lower than what was found in other evaluations in developed countries or reported by the manufacturer. This assay has a higher limit of detection when compared to Panbio and LumiraDx, which could justify the lower sensitivity observed in this evaluation. Differences in study settings, population immunity, epidemiological context, and workload at health facilities could have caused some of the differences in assay performance observed in our study. Evaluations performed in clinical settings and populations from Canada, United Kingdom, United States, and Germany reported discrepant sensitivities [24,25,26,27]. For example, in United Kingdom, in ambulance services and walk-through testing settings, Cubas-Attienzar et al. (2021) evaluated the COVIOS nasal antigen SARS-CoV-2 rapid tests and found a higher sensitivity and specificity, of 85.0% and 97.8%, respectively [26].

The genetic variants of SARS-CoV-2 prevalent at the time of the study are also a factor of possible importance in the performance of the evaluated assays. Our study was performed during high-transmission waves when the delta and omicron variants were predominant in Mozambique. Test manufactures have reported that emerging variants do not influence the performance of Ag-RDTs, because these detect nucleocapsid virus proteins while genetic variation of SARS-CoV-2 is mainly concentrated on the S gene [28]. Nevertheless, it has also been described that mutations affecting the spike protein can interfere with the binding of anti-nucleocapsid antibodies [28]. Although with no statistical significance, a study conducted by Bayart et al. (2022) found differences in Ag-RDTs sensitivity for delta and omicron variants, with sensitivity varying from 70.0 to 92.9% for delta and from 69.9 to 78.8% for omicron variant [29,30].

The nasal swab-based Ag-RDTs evaluated here had better performance than what we previously observed for nasopharyngeal swab-based Ag-RDTs [31]. Our results suggest that the simpler procedure of specimen collection positively impacts the clinical performance of assays. Additionally, our findings support the notion that self-sampling approaches for nasal swabs are feasible in LMICs, which could potentially increase access to SARS-CoV-2 testing. High-income countries have been successfully implementing self-testing in schools, offices, and universities [32]. Goggolidou et al. (2021) found that 79% of interviewers reported the willingness to perform home-based self-testing using easy-to-sample biological specimens [33]. The acceptability of self-collected nasal Ag-RDTs needs further investigation in the context of LMICs.

Primary healthcare is the main entry point to the health system for an estimated 80% of patients in LMICs [6]. The effective scaling-up of testing, for both endemic and epidemic diseases, can only be attained if using diagnostic procedures that are simple and do not require specialized biosafety conditions. Nasal swab-based rapid assays constitute a feasible and accurate alternative to more complex sampling and testing methods, especially in resource-poor settings.

## Figures and Tables

**Figure 1 biomedicines-10-02327-f001:**
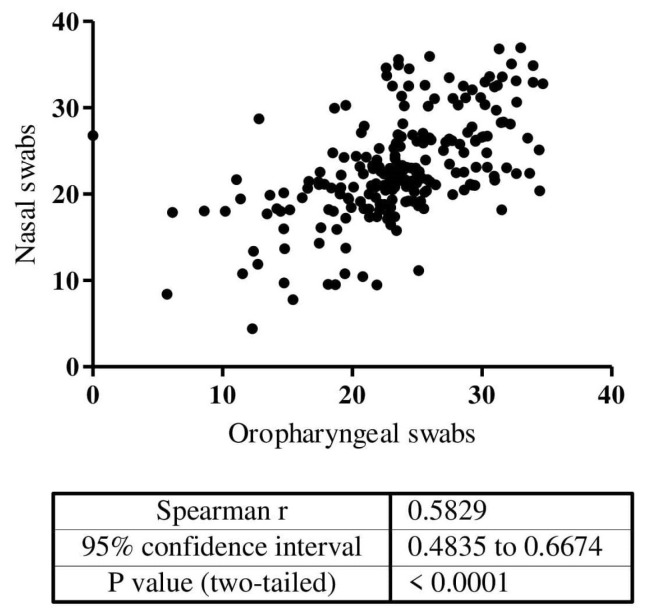
Correlation of Ct value of the positive cases (*n* = 213) in paired self-collected SteriPack nasal and oropharyngeal swab evaluation.

**Table 1 biomedicines-10-02327-t001:** Demographic characteristics of study participants.

		Nasal Swabs Self-Collection ^1^		Abbott Panbio ^2^		Covios ^3^		Lumira Dx 4	
		N = 1498	%	N = 400	%	N = 1123	%	N = 696	%
Gold standard ^5^ (RT-PCR or OP ^6^ swabs)	Negative	1229	82.04	186	46.5	586	53.0	310	44.54
	Positive	269	17.96	214	53.5	520	47.0	386	55.46
	Invalid	-- ^7^	--	--	--	--	--	--	--
	Indeterminate	--	--	--	--	--	--	--	--
Technology under evaluation ^8^	Negative	1227	81.91	228	57.0	773	68.8	379	54.45
	Positive	260	17.36	172	43.0	346	30.8	314	45.11
	Invalid	--	--	--	--	4	0.4	3	0.43
	Indeterminate	11	0.73	--	--	--	--	--	--
Study sites	CSM ^9^	200	13.35	--	--	--	--	256	36.78
	HGM ^10^	225	15.02	--	--	343	30.5	300	43.10
	HGC ^11^	384	25.63	400	100	395	35.2	--	--
	HPM ^12^	577	38.52	--	--	385	34.3	140	20.11
	HCM ^13^	112	7.48	--	--	--	--	--	--
Age	Median in years	35.0	--	38.02	--	33.0	--	36.23	--
Sex	Female	764	52.04	228	57.0	644	57.4	381	54.74
	Male	734	48.9	172	43.0	479	42.6	315	45.26
Symptoms	Yes	1356	90.52	397	99.25	1115	99.3	575	82.61
	No	141	9.48	3	0.75	8	0.7	121	17.39
Onset symptoms	≤7 days	1168	86.13	369	92.95	1033	92.0	519	90.26
	>7 days	188	13.87	28	7.05	82	7.3	56	9.74
Ct Value of PCR in OP samples, median (min-max)		24.7 (5.7–36.5)	NA ^14^	26.7 (9.0–37.3)	NA	25.2 (5.1–39.0)	NA	25.1 (3.7–35.9)	NA

^1^ SteriPack Sterile Polyester Nasal Swabs, ^2^ Panbio^TM^ COVID-19 Ag rapid diagnostic test device nasal, ^3^ LumiraDx SARS-CoV-2 Ag Test, ^4^ COVIOS^®^ Ag COVID-19 Rapid Antigen Test, ^5^ RT-PCR was gold standard for Ag-RDTs and Oropharyngeal swabs was gold standard for self-collected nasal swabs, ^6^ Oropharyngeal swabs, ^7^ Not available, ^8^ Technologies under evaluation are self-collected nasal swabs and Ag-RDTs., ^9^ Centro de Saúde de Marracuene, ^10^ Hospital Geral de Mavalane, ^11^ Hospital Geral de Chamanculo, ^12^ Hospital Provincial da Matola, ^13^ Hospital Central de Maputo. ^14^ Not Applicable.

**Table 2 biomedicines-10-02327-t002:** Performance of a self-collected nasal swab and three nasal swab-based antigen rapid tests for SARS-CoV-2 detection.

Diagnostic Test	Patient Type	Sensitivity	Specificity	PPV	NPV	Observed Agreement	Cohen’s Kappa
Self-collected nasal swab	Overall N = 1498	80.6% [75.3–85.2%]	96.4% [95.2–97.4%]	83.1% [78.0–87.4%]	95.8% [94.5–96.8%]	93.5% [34.7–94.7%]	0.78 [0.77–0.79]
	Symptomatic N = 1356	81.0% [75.5–85.7%]	96.7% [95.5–97.7%]	84.5% [79.2–88.9%]	95.9% [94.5–97.0%]	93.9% [34.7–95.1%]	0.79 [0.78–0.79]
	Asymptomatic N = 141	90.5% [69.6–98.8%]	93.1% [86.9–97.0%]	70.4% [49.8–86.2%]	98.2% [93.8–99.8%]	92.7% [31.0–96.4%]	0.75 [0.67–0.83]
	Symptoms ≤ 7 d ^1^ N = 1168	81.1% [75.2–86.2%]	96.5% [95.1–97.6%]	83.9% [78.1–88.7%]	95.8% [94.3–97.0%]	93.7% [34.6–95.0%]	0.79 [0.78–0.80]
	Symptoms > 7 d N = 188	93.3% [68.1–99.8]	98.1% [93.4–99.8%]	87.5% [61.7–98.4%]	99.0% [94.8–100.0%]	97.5% [30.7–99.5%]	0.89 [0.78–0.99]
Abbott Panbio Nasal RDT	Overall N = 400	79.7% [73.3–85.1%]	97.2% [93.5–99.1%]	96.8% [92.8–99.0%]	81.4% [75.5–86.4%]	88.0% [41.9–91.2%]	0.71 [0.70–0.72]
	Symptomatic N = 397	79.9% [73.3–85.1%]	97.1% [93.4–99.1%]	96.8% [92.8–99.0%]	81.2% [75.2–86.2%]	87.9% [42.1–91.1%]	0.71 [0.70–0.72]
	Asymptomatic N = 3	NA	NA	NA	NA	NA	NA
	Symptoms ≤ 7 d N = 369	80.0% [73.3–85.7%]	96.9% [93.0–99.0%]	96.6% [92.1–98.9%]	81.9% [75.7–87.0%]	88.2% [41.7–91.4%]	0.71 [0.70–0.72]
	Symptoms > 7 d N = 28	76.5% [50.1–93.1%]	100.0% [69.2–NA]	100.0% [75.3–NA]	71.4% [41.9–91.6%]	85.2% [34.1–95.8%]	0.72 [0.59–0.85]
	Overall N= 1102	59.6% [55.2–63.8%]	94.9% [92.8–96.5%]	91.1% [87.6–93.9%]	72.6% [69.3–75.8%]	78.3% [37.0–80.7%]	0.56 [0.551–0.560]
	Symptomatic N= 1094	59.7% [55.3–64.0%]	94.8% [92.7–96.5%]	91.1% [87.5–93.9%]	72.7% [69.3–75.8%]	78.3% [37.0–80.7%]	0.56 [0.552–0.561]
COVIOS Ag COVID-19	Asymptomatic N= 8	33.3% [0.8–90.6%]	100.0% [47.8–NA]	100.0% [2.5–NA]	71.4% [29.0–96.3%]	75.0% [13.3–96.8%]	0.38 [-0.628–1.397]
	Symptoms ≤ 7 d N= 1013	60.9% [56.5–65.2%]	94.6% [92.2–96.4%]	91.6% [88.0–94.3%]	71.4% [67.8–74.7%]	78.0% [37.6–80.5%]	0.56 [0.553–0.562]
	Symptoms > 7 d N= 81	20.0% [4.3–48.1%]	97.0% [89.5–99.6%]	60.0% [14.7–94.7%]	84.2% [74.0–91.6%]	82.7% [25.8–90.2%]	0.23 [-0.156–0.613]
LumiraDx SARS-CoV-2 Ag Test	Overall N = 696	78.0% [73.5–82.0%]	95.8% [92.9–97.7%]	95.9% [93.0–97.8%]	77.6% [73.0–81.7%]	85.9% [43.7–88.4%]	0.72 [0.72–0.73]
	Symptomatic N = 575	79.9% [75.4–83.9%]	95.7% [92.0–98.0%]	97.0% [94.4–98.6%]	73.4% [67.7–78.5%]	85.7% [47.2–88.5%]	0.71 [0.70–0.72]
	Asymptomatic N = 121	47.8% [26.8–69.4%]	95.9% [89.8–98.9%]	73.3% [44.9–92.2%]	88.6% [80.9–94.0%]	86.7% [28.7–92.2%]	0.50 [0.36–0.65]
	Symptoms ≤ 7 d N = 519	81.6% [77.0–85.6%]	96.2% [92.4–98.5%]	97.5% [94.9–99.0%]	74.6% [68.6–80.0%]	86.8% [47.9–89.6%]	0.73 [0.72–0.74]
	Symptoms > 7 d N = 56	62.5% [43.7–78.9%]	91.7% [73.0–99.0%]	90.9% [70.8–98.9%]	64.7% [46.5–80.3%]	75.0% [32.2–85.6%]	0.52 [0.44–0.59]

^1^ days.

## Data Availability

MDPI Research Data Policies.

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
