# Peer review of "Clinical Performance of Self-Collected Nasal Swabs and Antigen Rapid Tests for SARS-CoV-2 Detection in Resource-Poor Settings"

_biomedicines, 2022, doi:10.3390/biomedicines10092327_

Round 1
Reviewer 1 Report
Dear Authors,
I report on the review of your manuscript on clinical performance of self-collected nasa swabs and RATs for SARS-CoV-2.
The study design is appropriate and clear and the results are presented also clear and easy to understand.
Please revise a few sections in your manuscript:
1. I do not understand the lines "Gold standard (PCR or OP swab)" and "Technology under evaluation" in Table 1. Do you mean in the first line OP collection and analysis by PCR and in the second line the self-collected nasal samples? If so, please clarify and include lines "invalid" and "indeterminate" also in the "Gold standard" line, even though there are no such cases to report.
2. please specify for all test systems evaluated, whether nasal samples would comply with manufacturers' specifications on specimen to be used. If this does not meet manufacturers' specifications, please discuss the benefit of using the assays for off-label specimen and the need for validation of this deviating use.
3. Please include the limits of detection of all test systems used (incl. Abbott PCR) according to manufacturers' specifications and discuss whether the specifications are met.
4. some references got mixed up; e.g., in Discussion line 14 you refer to WHO requirements and refer to reference no. 20, but no. 20 is some other publication; WHO tech specs are listed as reference no. 21. Please verify the appropriateness of all references.
I will be happy to review your manuscript again after revision.
Best regards!
Author Response
- I do not understand the lines "Gold standard (PCR or OP swab)" and "Technology under evaluation" in Table 1. Do you mean in the first line OP collection and analysis by PCR and in the second line the self-collected nasal samples?
R: The gold standard was (1) RT-PCR for the evaluation of Ag-RDTs and (2) OP swabs for the evaluation of self-collected nasal swabs. It means that the results on gold standard lines referred to obtained by RT-PCR during the evaluation of Ag-RDTs or using OP swabs during the evaluation of self-collected nasal swabs. Please, see the table 1.
- If so, please clarify and include lines "invalid" and "indeterminate" also in the "Gold standard" line, even though there are no such cases to report.
R: We added invalid and indeterminate lines, as suggested. Please, see the table 1.
- please specify for all test systems evaluated, whether nasal samples would comply with manufacturers' specifications on specimen to be used. If this does not meet manufacturers' specifications, please discuss the benefit of using the assays for off-label specimen and the need for validation of this deviating use.
R: In the 2.4 Antigen Rapid Test (Ag-RDT) testing, we described in the first paragraph that the testing was performed accordingly to manufacturer instructions.
- Please include the limits of detection of all test systems used (incl. Abbott PCR) according to manufacturers' specifications and discuss whether the specifications are met.
R: We added the limit of detection for each Ag-RDT and PCR in the descriptions of the tests. We also discussed if the Ag-RDTs met the manufacturer specifications in terms of performance.
- some references got mixed up; e.g., in Discussion line 14 you refer to WHO requirements and refer to reference no. 20, but no. 20 is some other publication; WHO tech specs are listed as reference no. 21. Please verify the appropriateness of all references.
R: We reviewed the references, as suggested.
Reviewer 2 Report
Comments about the manuscript entitled:” Clinical performance of self-collected nasal swabs and antigen rapid tests for SARS-CoV-2 detection in re-source-poor settings”
General comment:
Although too much manuscripts are published concerning detection of COVID-19, but still any data based on clinical trials can help for better understanding of variations and thus handling this pandemic correctly.
1In discussion part: “Our findings on these two tests are similar to those found in a study in Germany [24].”: It is better that authors mention what specifically the findings that matched their results in reference 24 in more details.
2 Discussion in general needs more deep interpretation of results compared to findings from literature.
Author Response
- In discussion part: “Our findings on these two tests are similar to those found in a study in Germany [24].”: It is better that authors mention what specifically the findings that matched their results in reference 24 in more details.
R: We reviewed as suggested.
- Discussion in general needs more deep interpretation of results compared to findings from literature.
R: We reviewed all discussion, as suggested.
Round 2
Reviewer 1 Report
Dear authors,
thank you responding to my notes and for adapting the manuscript accordingly.
I'm happy with this manuscript now, congratulations, I will recommend acceptance for publication.
Best wishes